# Beneficial Effects of Linseed Supplementation on Gut Mucosa-Associated Microbiota in a Physically Active Mouse Model of Crohn’s Disease

**DOI:** 10.3390/ijms23115891

**Published:** 2022-05-24

**Authors:** Claire Plissonneau, Adeline Sivignon, Benoit Chassaing, Frederic Capel, Vincent Martin, Monique Etienne, Ivan Wawrzyniak, Pierre Chausse, Frederic Dutheil, Guillaume Mairesse, Guillaume Chesneau, Nathalie Boisseau, Nicolas Barnich

**Affiliations:** 1Microbes, Intestin, Inflammation et Susceptibilité de l’Hôte (M2iSH), UMR 1071 Inserm, USC-INRAE 2018, Université Clermont Auvergne, CRNH Auvergne, 63000 Clermont-Ferrand, France; claire.plissonneau@uca.fr; 2Laboratoire des Adaptations Métaboliques à l’Exercice en Conditions Physiologiques et Pathologiques (AME2P), Université Clermont Auvergne, CRNH Auvergne, 63000 Clermont-Ferrand, France; adeline.sivignon@uca.fr (A.S.); vincent.martin@uca.fr (V.M.); monique.etienne@uca.fr (M.E.); nathalie.boisseau@uca.fr (N.B.); 3Inserm U1016, Team ‘‘Mucosal Microbiota in Chronic Inflammatory Diseases’’, CNRS UMR 8104, Université de Paris, 75014 Paris, France; benoit.chassaing@inserm.fr; 4Unité de Nutrition Humaine (UNH), INRAE, Université Clermont Auvergne, CRNH Auvergne, 63000 Clermont-Ferrand, France; frederic.capel@inrae.fr; 5Institut Universitaire de France, 75005 Paris, France; 6Laboratoire Microorganismes, Génome et Environnement (LMGE), CNRS, Université Clermont Auvergne, 63000 Clermont-Ferrand, France; ivan.wawrzyniak@uca.fr; 7Laboratoire de Psychologie Sociale et Cognitive (LaPSCo), CNRS UMR 6024, Physiological and Psychosocial Stress, University Hospital of Clermont-Ferrand, Université Clermont Auvergne, 63000 Clermont-Ferrand, France; pierre.chausse@uca.fr (P.C.); frederic.dutheil@uca.fr (F.D.); 8CHU Clermont-Ferrand, Occupational and Environmental Medicine, WittyFit, 63000 Clermont-Ferrand, France; 9Valorex, La Messayais, 35210 Combourtillé, France; g.mairesse@valorex.com (G.M.); g.chesneau@valorex.com (G.C.)

**Keywords:** linseed, mucosa-associated microbiota, adherent-invasive *E. coli*, butyrate, inflammation

## Abstract

The Western diet, rich in lipids and in n-6 polyunsaturated fatty acids (PUFAs), favors gut dysbiosis observed in Crohn’s disease (CD). The aim of this study was to assess the effects of rebalancing the n-6/n-3 PUFA ratio in CEABAC10 transgenic mice that mimic CD. Mice in individual cages with running wheels were randomized in three diet groups for 12 weeks: high-fat diet (HFD), HFD + linseed oil (HFD-LS-O) and HFD + extruded linseed (HFD-LS-E). Then, they were orally challenged once with the Adherent-Invasive *Escherichia coli* (AIEC) LF82 pathobiont. After 12 weeks of diet, total energy intake, body composition, and intestinal permeability were not different between groups. After the AIEC-induced intestinal inflammation, fecal lipocalin-2 concentration was lower at day 6 in n-3 PUFAs supplementation groups (HFD-LS-O and HFD-LS-E) compared to HFD. Analysis of the mucosa-associated microbiota showed that the abundance of *Prevotella*, *Paraprevotella*, *Ruminococcus*, and *Clostridiales* was higher in the HFD-LS-E group. Butyrate levels were higher in the HFD-LS-E group and correlated with the Firmicutes/Proteobacteria ratio. This study demonstrates that extruded linseed supplementation had a beneficial health effect in a physically active mouse model of CD susceptibility. Additional studies are required to better decipher the matrix influence in the linseed supplementation effect.

## 1. Introduction

Crohn’s disease (CD) is an inflammatory bowel disease. Although the cause is unknown, many factors have been implicated in this pathology, such as genetic predisposition, environmental factors, immune system deregulation, and intestinal microbiota alterations [1]. Indeed, in patients with CD, bacterial diversity is reduced: members of the Firmicutes phylum, such as *Faecalibacterium prausnitzii*, are decreased, whereas members of the Bacteroidetes and Proteobacteria phyla, especially *Enterobacteriaceae* among which is *Escherichia coli* [2,3,4,5,6], are increased. However, it is not clear whether this dysbiosis is a cause and/or a consequence of CD. At the subspecies level, adherent-invasive *E. coli* (AIEC) has been detected more frequently in ileal samples from patients with CD than in healthy controls [7]. AIEC can adhere to and invade intestinal epithelial cells, and they also survive and replicate within macrophages [7,8]. Studies in CD animal models suggest an important role of AIEC in gut inflammation induction and/or maintenance [9,10,11]. AIEC strains bind to mannosylated receptors, such as carcinoembryonic antigen-related cell adhesion molecule 6 (CEACAM6) that is abnormally expressed at the enterocyte surface in patients with CD [12]. AIEC colonization is CEACAM6- and type 1 pili-dependent. Oral challenge with AIEC of CEABAC10 transgenic mice, that express the human CEACAM6, induced weight loss, diarrhea, rectal bleeding, and gut inflammation [9].

In CEABAC10 transgenic mice, the Western diet, which is considered a risk factor for CD development [13], increases AIEC capacity to colonize the gut mucosa, and then to trigger inflammation [14,15]. The Western diet is poor in fibers and rich in total fat, especially saturated fatty acids and n-6 polyunsaturated fatty acids (PUFAs), and in refined sugars and animal proteins. Animal models of diet-induced obesity and metabolic disorders demonstrated that high fat/high sugar diets increase the total fat mass (especially visceral fat mass), enhance metabolic disorders, favor chronic low-grade inflammation, and alter gut microbiota composition [14,15,16,17]. Thus, environmental factors, including long-term lifestyle habits, which influence gut microbiota community and diversity [18,19], may trigger CD, and could be a relevant target in its management. In developed Western countries, n-3 PUFA intake is much lower than that of n-6 PUFAs [20,21]. However, n-3 PUFAs exert beneficial effects on body composition, lipid profile, and the inflammation state in metabolic diseases [22,23]. They can also improve the intestinal barrier function and integrity [24] and influence gut microbiota composition in these metabolic pathologies [25,26,27]. A supplementation of n-3 PUFAs and a rebalance of n-6 PUFA intake could be a relevant strategy to manage chronic inflammation and dysbiosis in the remission/inactive phases of CD. For decades, marine products have been mainly used to evaluate the impact of n-3 PUFA supplementation, but currently, linseed, rich in α-linolenic acid (ALA, C18:3 n-3 PUFAs), is increasingly studied [26]. Linseed is one of the richest plants in ALA, the precursor of n-3 PUFAs. Post-ingestion, ALA is converted, to a limited extent, in EPA and DHA. Linseed is also composed of lignans, especially secoisolariciresinol diglucoside (SDG), fibers, and other bioactive components, that are not found in marine products. In addition, linseed has been previously used in human trials and demonstrated some beneficial effects on human health [28]. Therefore, besides the beneficial effects of n-3 PUFA, linseed, a matrix with fatty acids, fibers, lignans, and other bioactive components, may have a therapeutic effect on human health and CD prevention [29,30].

Moreover, regular physical exercise could be a complementary therapy to the rebalance of the n-6/n-3 PUFA ratio due to its anti-inflammatory effects and its capacity to decrease total fat mass, including visceral adipose tissue [31,32,33]. Indeed, in patients with CD, the ratio of (intra)-abdominal subcutaneous adipose tissue to total adipose tissue is significantly higher than in healthy controls [34]. In CD, this visceral fat mass, called “creeping fat”, surrounds the small intestine and colon and is metabolically active [35,36]. It leads to the release of pro-inflammatory mediators, such as tumor necrosis factor alpha (TNF-α) and interleukine-6 (IL-6) [37,38]. Studies in animal models of intestinal inflammation highlighted the anti-inflammatory effect of voluntary exercise and spontaneous physical activity [39,40]. Physical activity also may prevent intestinal injury following bacterial infection and/or exposure to bacterial components [41], and intestinal damage caused by the Western diet [42,43]. 

Therefore, the aim of this study was to assess the preventive effect of two forms of linseed supplementation (oil and extruded linseed) by rebalancing the n-6/n-3 PUFA ratio in an active mouse model of CD susceptibility fed a high fat diet (HFD). The study focused on the effects on body composition, AIEC-induced intestinal inflammation, and mucosa-associated microbiota. Two forms of linseed supplementation were used to determine whether the matrix could modify the biological effects [44,45].

## 2. Results

### 2.1. Body Composition in an Active Mouse Model That Mimics CD Susceptibility

First, 8-week-old male mice were randomly separated in three groups that were fed a HFD (n = 16), HFD with linseed oil (HFD-LS-O; n = 16), or HFD with extruded linseed (HFD-LS-E; n = 16) for 12 weeks. All mice had a running wheel in their individual cage and their spontaneous physical activity was continuously monitored.

Food intake (g and Kcal) and spontaneous physical activity (average distance per day) were not different in the three groups during the 12 weeks (Figure 1a,b), as well as body weight changes and weight gain (assessed as fat mass and fat-free mass) (Figure 1c,d). Similarly, no difference was observed in the percentage of mesenteric and epididymal adipose tissues relative to the total fat mass (Figure 1e,f). Nevertheless, the linoleic acid (C18:2 n-6)/α-linolenic acid (C18:3 n-3) (LA/ALA) ratio in the mesenteric adipose tissue was significantly decreased in both linseed supplementation groups (HFD-LS-O and HFD-LS-E) compared with the HFD group (Figure 1g).

### 2.2. AIEC-Induced Inflammation

After 12 weeks of diet, 1% dextran sulfate sodium (DSS) was added in the drinking water to damage the intestinal epithelium, followed (after 3 days) by the oral challenge with the AIEC strain LF82 (10^9^ bacteria/once). Weight monitoring at day 0 (just before the challenge) and at days 1, 3, and 6 after the challenge did not find any difference in the three groups, indicating that the linseed supplementation did not influence AIEC-induced weight loss (Figure 2a). Quantification of fecal lipocalin-2 by ELISA, a non-invasive, sensitive, dynamic, stable, and cost-effective biomarker to monitor intestinal inflammation in mice [46], did not show any significant difference in the HFD and HFD-LS-O/HFD-LS-E groups at day 1 post-challenge, indicating a similar level of intestinal inflammation in the three groups at this time point. At day 6, fecal lipocalin-2 levels were significantly higher in the HFD group than in the HFD-LS-O/HFD-LS-E groups, suggesting that linseed supplementation (both forms) limited AIEC-induced intestinal inflammation (Figure 2b). Fecal lipocalin-2 levels at day 6 were similar in the HFD-LS-O (pink triangles) and HFD-LS-E (green circles) groups, indicating that the limiting effect was independent of the matrix. The negative correlation (*p* < 0.05) between fecal lipocalin-2 concentrations and level of physical activity (mean distance in km per day) (Figure 2c) suggested that physical activity may contribute to the limitations of AIEC-induced intestinal inflammation.

As intestinal inflammation can lead to increased intestinal permeability (leaky gut), which is a major problem in inflammatory bowel disease, intestinal permeability was evaluated in vivo by measuring FITC-dextran 4 kDa (exogenous molecule provided by oral gavage) in serum (at week 12) and lipopolysaccharides (LPS; bacterial component naturally present in the microbiota) in plasma (at day 7 post-challenge). FITC-dextran and LPS levels were comparable in the three groups (Figure 2d,e), indicating that the linseed supplementation did not have a major effect on the intestinal barrier function.

### 2.3. Intestinal Mucosa-Associated Microbiota Diversity and Composition

At day 7 after the AIEC challenge, analysis of the colon mucosa-associated microbiota composition by 16S rRNA sequencing showed that the α-rarefaction curve contained significantly more OTUs in the HFD-LS-E group (*p* < 0.05) (Figure 3a). This indicated that the addition of extruded linseed to the HFD diet had a beneficial effect on microbial species. Moreover, the number of observed OTUs was significantly increased in the HFD-LS-E group compared with the HFD and HFD-LS-O groups (Figure 3b). The α-diversity, represented by the Shannon index, tended to increase in the HFD-LS-E group (*p* = 0.10) (Figure 3c). Evenness was comparable among groups (Figure 3d). Conversely, the phylogenic diversity was significantly increased (*p* < 0.01) in the HFD-LS-E group compared with the HFD and HFD-LS-O groups (Figure 3e). The β-diversity analysis by principal coordinate analysis of the weighted and unweighted UniFrac distance matrices showed that colon microbiota composition was changed only when using unweighted UniFrac data in the HFD-LS-E group compared with the HFD and HFD-LS-O groups (*p* < 0.05) (Figure 3f,g).

Concerning the abundance of specific microbiota phyla in the colon mucosa, Firmicutes tended to increase (*p* = 0.064) in both supplemented groups (HFD-LS-O and HFD-LS-E) (Figure 4a). Compared with the HFD group, linseed supplementation increased the Firmicutes/Proteobacteria ratio (*p* = 0.03) (Figure 4b). The analysis of composition of microbiomes (ANCOM) highlighted significant group differences for specific mucosa-associated microbiota species (Figure 4c,f). Indeed, in the HFD-LS-E group, the relative abundance of *Clostridiales* spp., *Paraprevotella* spp., and *Prevotella* spp. was significantly increased compared with the HFD and HFD-LS-O groups (*p* < 0.05) (Figure 4c,d,f) and that of *Ruminococcus* spp. compared with the HFD group (*p* < 0.05) (Figure 4f). SCFAs produced by the microbiota were quantified in fecal samples. No difference was found among groups concerning acetate, butyrate, propionate, and total SCFAs concentrations (Table 1). Butyrate, a major SCFA (due to its fermentative activity and beneficial effects at the intestinal level), tended to increase only in the HFD-LS-E group (Figure 5a), in line with the observed increased percentage of SCFA-producing Firmicutes in this group. When only mice in which fecal butyrate could be detected were considered, their percentage was significantly higher in the HFD-LS-E group (*p* < 0.0001) (Figure 5b). Furthermore, the level of fecal butyrate was positively correlated (*p* = 0.0163) with the mucosa-associated Firmicutes/Proteobacteria ratio (Figure 5c).

## 3. Discussion

The aim of this study was to determine whether two different forms of linseed supplementation (oil and extruded seeds as specific matrix) to rebalance n-6/n-3 PUFA ratio in a high-fat diet have a preventive effect in active mice that mimic CD susceptibility. In our experimental conditions that included spontaneous physical activity, the two linseed forms seem to act beneficially on AIEC-induced intestinal inflammation, but only HFD-LS-E modulated the gut mucosa-associated microbiota diversity, increased the abundance of *Clostridiales*, *Prevotella*, and *Ruminococcus* bacteria, and enhanced the production of beneficial metabolites, such as fecal butyrate.

In the present study, we chose to use CEABAC10 transgenic mice fed a HFD and challenged with AIEC because we previously reported that this mouse model mimics what occurs in patients with CD [14,15]. Indeed, in these mice, HFD induces gut microbiota composition changes and alters the host homeostasis by promoting AIEC encroachment [14]. In addition, spontaneous physical activity in this model decreases total fat mass, improves glucose metabolism, and promotes healthy gut microbiota composition changes, with an increase in SCFA-producing species belonging to the *Oscillospira* and *Ruminococcus* genera, linked to higher fecal levels of propionate and butyrate [40]. The Western diet is characterized by low fiber and high fat intakes, especially n-6 PUFAs and saturated and trans fatty acids, and this could explain the increase in total fat mass and chronic low-grade inflammation, and the gut microbiota dysbiosis [20,47]. Therefore, we evaluated the impact of spontaneous physical activity level combined with n-6/n-3 PUFA ratio rebalancing (using linseed, rich in n-3 PUFAs) on gut mucosa-associated microbiota in HFD-fed CEABAC10 mice after AIEC-induced inflammation. Mice were challenged with the AIEC LF82 strain only once and no antibiotics were used in this study, to preserve the gut mucosa-associated microbiota. As previously reported, the level of colonization and the impact on gut function decreased one week post-infection [14,15]. Our results suggest that linseed supplementation (whatever the matrix) may limit AIEC-induced intestinal inflammation as shown by the lower fecal lipocalin-2 concentrations (a sensitive and non-invasive inflammation biomarker [46]) in the supplemented groups, without affecting the intestinal permeability. It is known that n-3 PUFAs have anti-inflammatory properties, particularly after stimuli that promote inflammation [48]. Indeed, after 12 weeks of supplementation, n-3 PUFAs should have been converted to eicosapentaenoic acid (EPA) and docosahexaenoic acid (DHA), the active metabolic n-3 PUFA derivatives. These derivatives should have been incorporated into the cell membrane phospholipids, to be released and converted into “specialized pro-resolving mediators”, in order to indirectly inhibit the NF-κB pathway, in response to acute inflammation [49], as the AIEC exposure in this study. We hypothesized that n-3 PUFA intake from linseed, independent of form intake (HFD-LS-O and HFD-LS-E groups), may have limited AIEC-induced inflammation. Long-term n-3 PUFA intake could be interesting for preventing acute intestinal inflammation, and for improving the resolution of acute inflammation. As CD is characterized by acute phases with inflammation, and periods of remission, our results also suggest that in the long term, n-6/n-3 PUFA rebalancing may decrease the susceptibility to relapse during re-exposure to AIEC and prolong periods of remission. 

As CD is associated with detrimental changes in the microbiota composition and function, we then evaluated the impact of the two linseed forms (HFD-LS-O and HFD-LS-E) on the intestinal mucosa-associated microbiota in active HFD-fed CEABAC10 mice. In addition, we recently demonstrated in a cohort of 102 patients with Crohn’s disease that the presence of the AIEC pathobiont in the mucosa, but not in stool, correlates with the inflammatory state [50]. Therefore, we decided to focus on the mucosa-associated microbiota rather than on the luminal and/or fecal microbiota. HFD-LS-E modulated β-diversity and tended to increase α-diversity (i.e., Shannon index). Moreover, OTUs were significantly increased in the HFD-LS-E group compared with the HFD and HFD-LS-O groups, demonstrating a greater impact of HFD-LS-E supplementation on the gut mucosa-associated microbiota diversity. Thus, extruded linseed, as a specific matrix that includes different components (e.g., lignans and especially SDG and other bioactive components [51]) had the most significant effect on the mucosa-associated microbiota compared. 

Interestingly, at the phylum level, the Firmicutes/Proteobacteria ratio was increased in both linseed-supplemented groups. This is in favor of a less colitogenic microbiota, because Firmicutes contain many beneficial SCFA-producing species, whereas Proteobacteria have pro-inflammatory properties [52,53,54]. The analysis of specific features of the mucosa-associated microbiota highlighted several differences in the three groups. First, *Clostridiales* spp. abundance was significantly increased in the HFD-LS-E group compared with the HFD and HFD-LS-O groups. Moreover, HFD-LS-E also induced an increase in the *Ruminococcus*, *Prevotella* and *Paraprevotella* genera. *Clostridiales* are increased in patients with obesity [55], but the influence of linseed on *Clostridiales* abondance is not known, and several studies even reported a *Clostridiales* decrease after fish oil supplementation [25,56,57]. Conversely, Zhu et al. showed an increase in *Clostridiales* in mice after gavage of EPA and DHA (60 mg). Therefore, *Clostridiales* abundance might be influenced by the n-3 PUFA type and dose [58]. Interestingly, *Paraprevotella* is a succinate producer and may play a role in energy metabolism. To our knowledge, only few studies assessed *Paraprevotella* abundance in humans associated with physical activity, n-3 PUFA intake, and linseed supplementation. *Paraprevotella* abundance might be positively associated with lower body mass index, lower Mediterranean diet adherence, and lower physical activity levels [59]. Moreover, in vitro experiments suggest that its abundance may be increased in sheep rumen after exposure to n-3-PUFAs (n-3 docosapentaenoic acid, EPA, and DHA) [60]. *Ruminococcus* is a genus that includes butyrate-producing species and is less abundant in patients with CD [61]. Li et al. reported the *Ruminococcus* genus increase after linseed oil supplementation [62]. This effect may be due to other linseed components than n-3 PUFAs. Indeed, linseed includes ~1% of SDG, phytoestrogens that are metabolized by the gut microbiota to secoisolariciresinol (SECO), and then to enterodiol and enterolactone that are bioactive polyphenolic components [63]. The gut microbiota diversity and composition seem to be linked to the production [64] and excretion of enterolactone and enterodiol [65,66]. The relationship between SDG and gut microbiota has been investigated mainly to evaluate the role of bacterial species in SDG conversion [64,67,68]. Some *Ruminococcus* species seem to play an important role in SDG biotransformation, such as *Ruminococcus gnavus* [44,69], *Ruminococcus bromii*, and *Ruminococcus lactaris* [64], or even *Ruminococcus albus* and *Ruminococcus flavefaciens* [68]. Fuentealba et al. compared the effect of linseed flour and whole linseed on the digestive process in vitro. They found that SCFA production was increased by linseed flour due to a greater fermentation process [70]. Thus, linseed flour may increase SDG release and enterodiol production compared with whole linseed. A recent study on the impact of whole linseed, linseed oil, or SDG supplementation in female mice [44] found that whole linseed is necessary for the complete SDG metabolization by the gut microbiota. Indeed, although the gut microbiota responds to specific dietary inputs (e.g., *R. gnavus* in an SDG-supplemented diet), the cooperative activities of a diverse microbiota are necessary to process linseed components. The authors also suggested that other unknown linseed components may be essential to optimize SDG digestion and conversion to enterodiol and enterolactone [44]. In 1996, Rickard et al. also observed in rats that SDG is not entirely metabolized, unlike whole linseed [45]. They hypothesized that another lignan precursor may exist in linseed. The *R. gnavus gus* gene encodes β-glucuronidase, an enzyme that plays a major role in the absorption and enterohepatic circulation of aglycone residues, such as lignans. Beaud et al. showed that SDG may stimulate *R. gnavus* growth by acting as a substrate [69]. Only one randomized, placebo-controlled cross-over study analyzed the human gut microbiota composition following enterolactone and enterodiol supplementation [66] and found that enterolactone and enterodiol were linked to the gut microbiota composition. Moreover, the excretion of enterolactone and enterodiol was positively correlated with *Ruminococcaceae* abundance [66]. The authors also found transcriptomic differences between low and high enterolactone excreters and suggested that enterolactone or other activities associated with the gut microbiota modulate the responses to a lignan-supplemented diet [66]. On the other hand, *R. gnavus* may be associated with CD by producing inflammatory polysaccharides [71]. More studies are needed to precisely determine the role of *Ruminococcus*, especially of *R. gnavus,* in CD.

We also found that in the HFD-LS-E group, the relative abundance of *Prevotella,* which is present in fiber-rich diets and carbohydrate-rich diets, was increased. The fibers and lignans contained in extruded linseed may have contributed to this increase. Schogor et al. showed that in vitro and in vivo, *Prevotella* spp. may contribute to SDG conversion to SECO in the rumen of animals [68]. Consistent with our data, Li et al. found an increase in *Prevotella* in the rumen of animals after a diet supplemented in linseed oil and propionate precursors [62]. The linseed supplementation, as a matrix, may have stimulated *Prevotella* spp. growth. We previously reported that *Prevotella* increases in obese-induced Wistar rats after a 12-week high-intensity interval training program [72]. Petersen et al. found that *Prevotella* abundance increase in the microbiota of competitive cyclists is correlated with the mean exercise time per week [73]. Conversely, here, we did not observe any correlation between mean spontaneous exercise distance and *Prevotella* abundance. This could be explained by the fact that mice performed spontaneous activity, and it has been reported that spontaneous and imposed exercise modulate differently the gut microbiota [74]. Thus, it could be relevant in the future to assess the effect on *Prevotella* abundance of a controlled exercise modality, such as moderate intensity continuous training [32] or high intensity interval training [72], combined with HFD-LS-E supplementation.

Our study shows that in active CEABAC10 mice challenged with AIEC to mimic CD susceptibility, rebalancing the n-6/n-3 PUFA ratio of the HFD by linseed supplementation, as a matrix, promotes colon mucosa-associated microbiota composition changes. The abundance of butyrate producers, such as the *Clostridiales* order and *Ruminococcus* genus that are considered health-related bacteria, was increased in the HFD-LS-E group. These findings suggest that in the long term, the microbiota modulations induced by extruded linseed supplementation may prevent the onset of chronic pathologies that involve dysbiosis in active individuals and may extend the remission period in patients with CD.

## 4. Materials and Methods

### 4.1. Animals

FVB/N female and heterozygous CEABAC10 transgenic male mice (Charles River Laboratories) were mated in specific pathogen-free conditions in the animal care facility of the University Clermont Auvergne (Clermont-Ferrand, France). At 4 weeks of age, animals were weaned and genotyped. Eight-week-old CEABAC10 males (n = 48) were selected and randomly assigned to the three groups: HFD (n = 16), HFD + linseed oil (HFD-LS-O; n = 16), and HFD + extruded linseed (HFD-LS-E; n = 16) (Figure 6). Animals were housed in individual cages with a reversed light–dark cycle in a temperature-controlled room (20 ± 2 °C). All mice had a running wheel in their individual cage and their spontaneous physical activity was continuously monitored. All animal procedures were approved by the local ethics committee (C2EA-02, Auvergne, France; APAFIS 19400-2019022211242750) and were performed in accordance with the European animal welfare regulations and guidelines (European Directive 2010/63/EU on the protection of vertebrate animals used for experimental and scientific purposes). All efforts were made to protect animal welfare and to minimize suffering at each step of the protocol.

### 4.2. Spontaneous Physical Activity

Mice (HFD, HFD-LS-E and HFD-LS-O groups) performed voluntary wheel running in their individual cage. To measure the covered distance, each wheel was monitored with a magnet and a sensor connected to a microcontroller digital input–output card (PIC18 4550 MICROCHIP). A card was programmed specifically for this use and allowed to record each passage of the magnet in front of the sensor. The information was sent back to a computer for processing and storage using a specific program written in G language (LABVIEW National Instrument). Data analysis with MATLAB (MathWorks^®^, Natick, MA, USA) allowed the calculation of the travelled distance (km). Exercise was recorded continuously, and cages were visually checked at least five times/week.

### 4.3. Diets

The diets were prepared by INRAE (Jouy-en-Josas, France). During the entire study, the HFD (i.e., a high fat/high sucrose diet as a paradigm of the Western diet) contained 43.3% of fat, 39.4% of carbohydrates, and 17.3% of proteins, for a total energy amount of 468.40 kcal/100 g and a n-6/n-3 PUFA ratio of 14.80 (Table 2). To prepare the oil or extruded linseed-supplemented diets, the n-6/n-3 PUFA ratio, the macronutrient percentage, and the total kcal/100 g were considered (Table 2). The diet of the HFD-LS-O group contained 41% of fat, 41% of carbohydrates, and 18% of proteins, for a total energy amount of 468.40 kcal/100 g and a n-6/n-3 PUFA ratio of 3.3. The diet of the HFD-LS-E group was similar, but included extruded linseed (Tradilin, Patent n°FR1760984). It contained 41% of fat, 41.7% of carbohydrates, and 18% of proteins, for a total energy amount of 460.44 kcal/100 g and a n-6/n-3 PUFA ratio of 3 (Table 2). The calculated diet compositions are listed in Table 2 and Table 3. Extruded linseed is a matrix that includes other components listed in Figure 7.

### 4.4. Study Design

Groups followed their own diet (HFD, HFD-LS-O, and HFD-LS-E) during the entire protocol. Pair feeding was performed throughout the protocol to ensure that groups consumed the same food quantity, as spontaneous physical activity can increase food intake [75]. Mice had access to food and drinking water *ad libitum*. After 12 weeks of protocol, dextran sulfate sodium (1%) (MP biomedicals, Irvin, CA, USA) was added to the drinking water. After three days of DSS, all mice were also orally challenged once with the AIEC LF82 strain (10^9^ bacteria), isolated from a patient with CD. The AIEC LF82 strain was grown in LB broth (Condalab, Madrid, Spain), without agitation, at 37 °C for 24 h. Seven days after the AIEC LF82 strain exposition, mice were euthanized by cervical dislocation (Figure 6).

### 4.5. Weight and Body Composition

Body composition (weight, fat mass, and free-fat mass) was evaluated using an EchoMRI device (EchoMRI Medical System, Houston, TX, USA) at weeks 1, 6, and 12 before the oral challenge with the AIEC LF82 strain. Body weight was measured three times per week using a standard scale during the first 12 weeks, and then on infection day and at days 1, 3, and 6 post-challenge. Mesenteric and epididymal total adipose tissues were weighted post-mortem (day 7 post-challenge).

### 4.6. Polyunsaturated Fatty Acids in Mesenteric Adipose Tissue

Mesenteric adipose tissue samples (15–20 mg) were homogenized in 2:1 (*v*/*v*) chloroform/methanol solution using ceramic beads and a Minilys system (Bertin instruments, Ozyme, Montigny-le-Bretonneux FRANCE) according to the method described by Folch et al. [76]. Total lipids were isolated for fatty acid methylation analysis as described previously [72] using gas chromatography and a Finnigan Trace GC Ultra instrument (Waltham, MA, USA).

### 4.7. Inflammatory Markers

Feces were collected at days 1, 3, and 6 post-AIEC challenge. Samples were weighed, and then homogenized (without marbles) in phosphate buffer saline (PBS) for 15 min before centrifugation at 12,000× *g* rpm, at 4 °C for 10 min. Lipocalin-2 was quantified with an ELISA kit (Biotechne, Minneapolis, MN, USA) in duplicate.

LPS activity in plasma samples was evaluated, in duplicate, using HEK-Blue-mTLR4 cells (InvivoGen, San Diego, CA, USA). Briefly, 180 µL of cell suspension (1.4 × 10^4^ cells per mL of HEK-Blue Detection medium) (InvivoGen, San Diego, CA, USA) was added to 20 µL of each diluted (1:10) plasma sample. LPS (InvivoGen, San Diego, CA, USA) was used as the positive control and to calculate the standard range. Plates were incubated at 37 °C in 5% CO_2_ for 24 h, and alkaline phosphatase activity was measured at 620 nm.

### 4.8. Fecal Short-Chain Fatty Acids

Weighted fecal samples were reconstituted in 200 µL MilliQ^®^ water, homogenized, incubated at 4 °C for 2 h, and then centrifuged at 12,000× *g* at 4 °C for 15 min. Supernatants were weighed, saturated phosphotungstic acid (100 µL for 1 g of fecal content) was added, and samples were incubated at 4 °C overnight. After centrifugation, short chain fatty acid (SCFA) concentrations (including butyrate, propionate, and acetate) were determined by gas chromatography (Nelson 1020, Perkin-Elmer, St. Quentin en Yvelines, France). Briefly, chromatographic separation was achieved on DB-FFAP columns (30 m × 250 μm, 0.25 μm). The injector temperature was 250 °C and the injection volume was 1 μL. The initial oven temperature was 100 °C, and then was increased to 250 °C (10 °C/min) and held for 5 min. The carrier gas was helium at a constant flow of 7 mL/min. Samples were injected using the split mode at a ratio of 10:1. Detection was performed with a flame ionization detector (FID).

### 4.9. Intestinal Permeability: FITC-Dextran

In vivo intestinal permeability was measured using FITC-dextran 4 kDa (FD4, Sigma, St. Louis, MO, USA) at week 12 of the diet. Mice were orally challenged with 7.5 mg FD4 diluted in PBS for 5 h before blood collection. Serum was separated by centrifugation (5000× *g*, 30 min) and FITC concentration was determined by fluorescence measurement. 

### 4.10. Microbiota Composition Analyses

For genomic DNA extraction, mouse colon samples were lysed in proteinase K at 56 °C in a shaking incubator overnight. DNA was extracted using the Nucleospin^®^ Tissue kit (Macherey-Nagel, Germany). DNA concentration was determined with a QubitTM fluorometer (Invitrogen), and the DNA quality was evaluated with a NanoDrop™ (Thermo Scientific) spectrophotometer (260/280 and 260/230 ratio). Region V4 of the 16S rRNA gene was PCR-amplified using forward and reverse primers that were designed with the Golay error-correcting scheme and used to tag PCR products from individual samples [77]. The sequence of the composite forward primer 515F was: 5′-*AATGATACGGCGACCACCGAGATCTACACGCT*XXXXXXXXXXXX**TATGGTAATT*GT***GTGYCAGCMGCCGCGGTAA-3′. The italicized sequence represents the 5′ Illumina adapter, the 12X sequence is the Golay barcode, the bold sequence is the primer pad, the italicized and bold nucleotide is the primer linker, and the underlined sequence is the conserved bacterial sequence. The sequence of the reverse primer 806R was: 5′-*CAAGCAGAAGACGGCATACGAGAT***AGTCAGCCAG*CC***GGACTACNVGGGTWTCTAAT-3′. The italicized sequence is the 3′ reverse complement sequence of the Illumina adapter, the bold sequence is the primer pad, the italicized and bold nucleotides are the primer linker, and the underlined sequence is the conserved bacterial sequence. PCR mixtures included the Hot Master PCR mix (Quantabio, Beverly, MA, USA), 0.2 µM of each primer, and 10–100 ng of template. The reaction conditions were: 3 min at 95 °C, followed by 30 cycles of 45 s at 95 °C, 60 s at 50 °C, and 90 s at 72 °C on a BioRad thermocycler. PCR products were purified with Ampure magnetic purification beads (Agencourt, Brea, CA, USA) and visualized by gel electrophoresis. Products were then quantified (BIOTEK fluorescence spectrophotometer) using the Quant-iT PicoGreen dsDNA assay. A master DNA pool was generated from the purified products mixed in equimolar ratios. The pooled products were quantified using the Quant-iT PicoGreen dsDNA assay and then sequenced using an Illumina MiSeq sequencer (paired-end reads, 2 × 250 bp) at Cornell University, Ithaca.

The 16S rRNA sequences were analyzed using QIIME2—version 2019 [78]. Sequences were demultiplexed and quality filtered using the Dada2 method [79] with QIIME2 default parameters in order to detect and correct Illumina amplicon sequence data, and a table of Qiime 2 sequence variants (SVs) was generated. A tree was next generated, using the align-to-tree-mafft-fasttree command, for phylogenetic diversity analyses, and alpha and beta diversity analyses were computed using the core-metrics-phylogenetic command. Principal coordinate analysis (PCoA) plots were used to assess the variation between the experimental groups (beta diversity). For taxonomy analysis, features were assigned to operational taxonomic units (OTUs) with a 99% threshold of pairwise identity to the Greengenes reference database 13_8 [80]. All sequencing raw data have been deposited in European Nucleotide Archive (ENA) under accession number PRJEB48648.

### 4.11. Statistical Analysis

All statistical analyses were performed with the Statistica software (version 12). Data were presented as the mean ± standard deviation (SD). Normal data distribution was tested using the Kolmogorov–Smirnov test, and the homogeneity of variance with the F-test. Then, a *t*-test or a Mann–Whitney test was performed to compare the two supplemented groups (HFD-LS-O + HFD-LS-E pooled together) vs. HFD. A one-way ANOVA or a Kruskal–Wallis test was used to compare post-mortem values in the three groups. One-way ANOVA with repeated measures was used to determine group (G) effect, time (T) effect, and T*G interactions. For repeated measures, in the absence of normal distribution or variance homoscedasticity, data were first log transformed. The Newman–Keuls post-hoc test was used for all ANOVA analyses. Categorical data were compared with the Chi-square test. Finally, Spearman correlations were used to test relationships between variables. Differences were considered significant when *p*-values < 0.05.

## Figures and Tables

**Figure 1 ijms-23-05891-f001:**
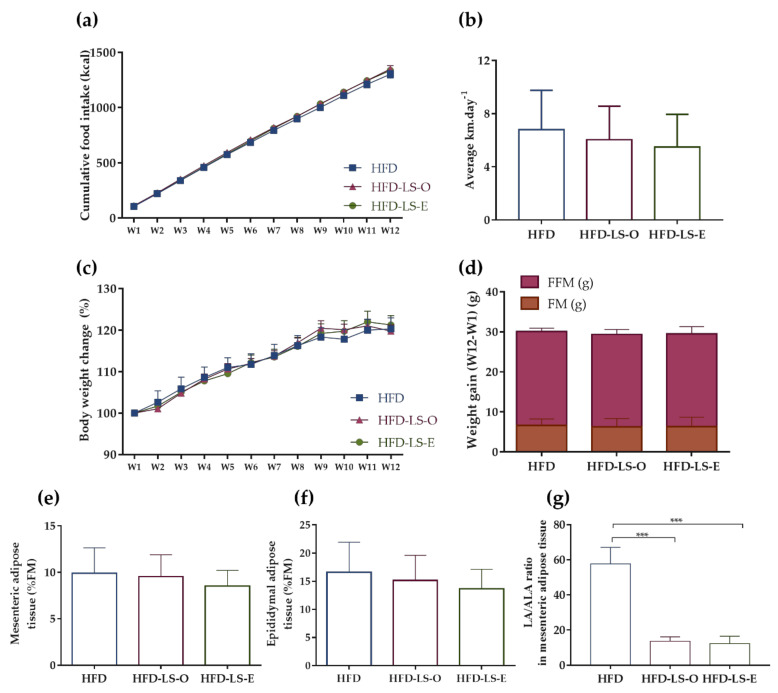
Effect of 12 weeks of voluntary physical activity and HFD diet, supplemented or not with linseed, on the cumulative food intake (kcal) (**a**), mean distance covered by wheel running (km·day^−1^) (**b**), body weight changes (%) (**c**), weight gain (week 12—week 1): fat mass (FM) (g) and fat-free mass (FFM) (g) (**d**), mesenteric adipose tissue (%FM) (**e**), epididymal adipose tissue (%FM) (**f**), and LA/ALA ratio in mesenteric adipose tissue (**g**) in the three groups. Data are the mean ± SEM (**a**,**c**) or mean ± SD (**b**,**d**–**g**). LA: linoleic acid; ALA: α-linolenic acid. *** *p* < 0.001.

**Figure 2 ijms-23-05891-f002:**
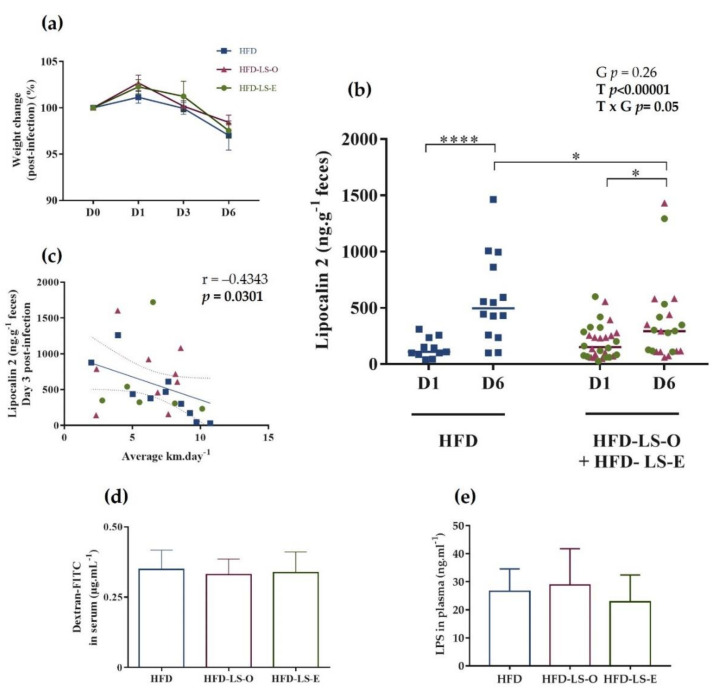
Weight change (%) after the oral challenge with the AIEC strain LF82 (**a**). Fecal lipocalin-2 concentration (µg·g^−1^ feces) at day 1 (D1) and D6 post-challenge (**b**), and correlation between lipocalin-2 concentration (µg·g^−1^ feces) at D3 post-challenge and mean wheel running distance (km) per day (**c**) in the three groups. Intestinal permeability was assessed by measuring FITC-dextran 4 kDa in the serum 5 h after intragastric administration of 7.5 mg FITC-dextran (*n =* 16 mice/group) at week 12, before the AIEC challenge (**d**). Plasma concentration (ng·mL^−1^) of active LPS 7 days post-AIEC pathobiont challenge (**e**). HFD-LS-O: pink triangles; HFD-LS-E green circles. * *p* < 0.05, **** *p* < 0.0001.

**Figure 3 ijms-23-05891-f003:**
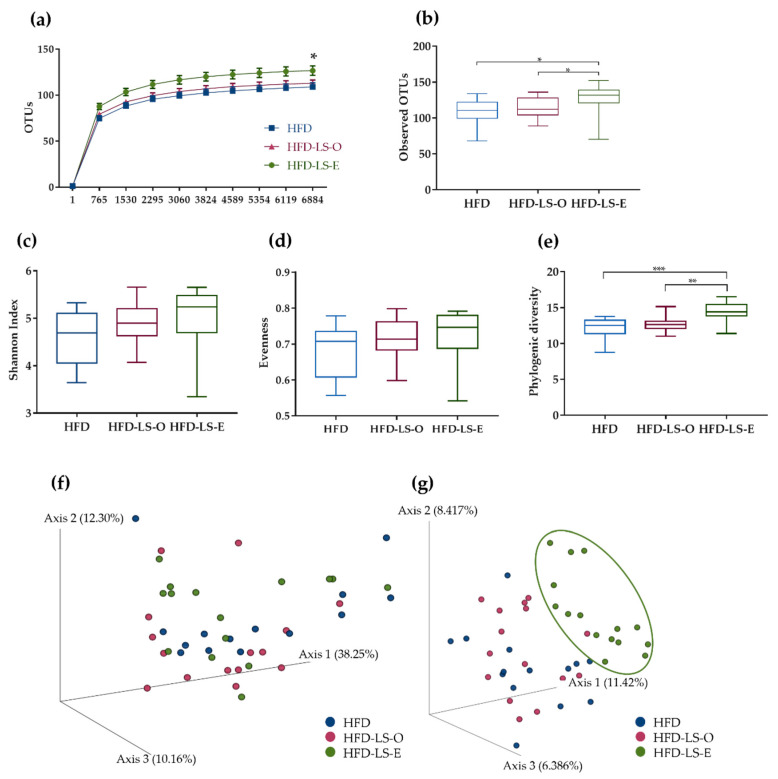
Mucosa-associated microbiota composition analyzed by 16S rRNA gene sequencing in DNA samples (Illumina MiSeq system) from HFD (n = 14), HFD-LS-O (n = 15) and HFD-LS-E (n = 15) colon samples collected at day 7 after oral challenge with the AIEC strain LF82. Operational taxonomic units (OTUs) with a rarefaction depth of 6884 sequences (**a**), observed OTUs (**b**), Shannon index (**c**), evenness (**d**), phylogenic diversity (**e**), weighted (**f**), and unweighted Unifrac analysis (**g**). * *p* < 0.05, ** *p* < 0.01, and *** *p* < 0.001.

**Figure 4 ijms-23-05891-f004:**
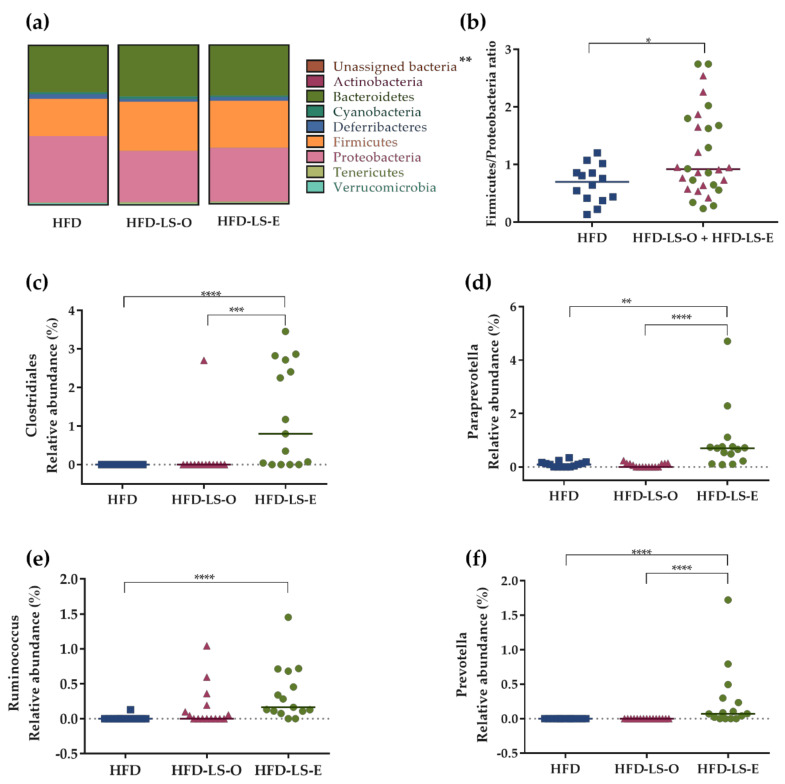
Composition of the mucosa-associated microbiota. Phylum distribution (%) (**a**), Firmicutes/Proteobacteria ratio (**b**), and relative abundance (%) of specific bacterial species in the mucosa-associated microbiota: *Clostridiales* (**c**)*, Paraprevotella* spp. (**d**), *Ruminococcus* (**e**), and *Prevotella* (**f**). Colon samples from HFD (*n* = 14), HFD-LS-O (*n* = 15), and HFD-LS-E (*n* = 15) mice collected at day 7 post-challenge with the AIEC strain LF82. * *p* < 0.05, ** *p* < 0.01, *** *p* < 0.001, **** *p* < 0.0001.

**Figure 5 ijms-23-05891-f005:**
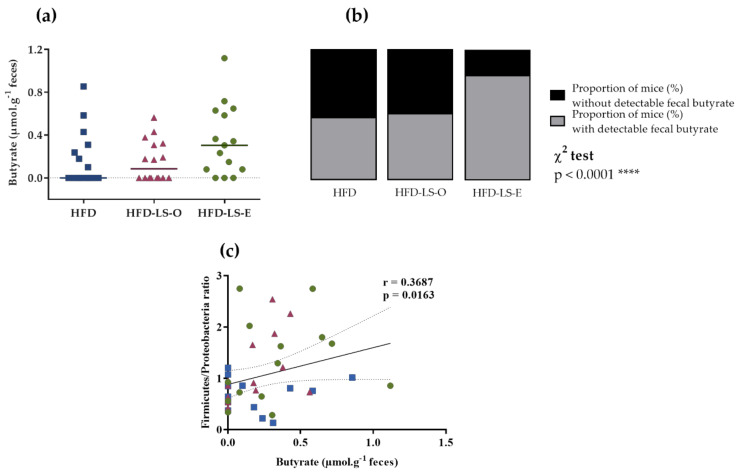
Fecal butyrate concentration (µmol·g^−1^) (**a**), percentage of mice with detectable fecal butyrate in each group (**b**), and correlation between fecal butyrate concentration and Firmicutes/Proteobacteria ratio (**c**). HFD (*n* = 14), HFD-LS-O (*n* = 15), and HFD-LS-E (*n* = 15). **** *p* < 0.0001.

**Figure 6 ijms-23-05891-f006:**
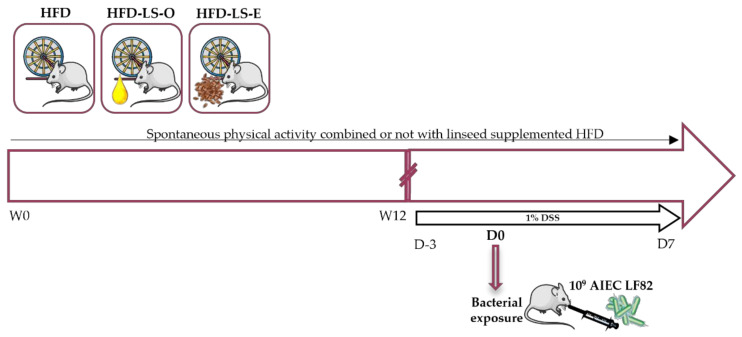
Study protocol. Eight-week-old mice were randomly divided in three groups that performed voluntary wheel running: HFD (high-fat diet, n = 16), HFD-LS-O (linseed oil-supplemented-HFD, n = 16), and HFD-LS-E (extruded linseed-supplemented-HFD, n = 16). The distance covered by wheel running was recorded continuously for 12 weeks. At the end of week 12 (W12), 1% dextran sulfate sodium (DSS) was added in the water of mice and after 3 days, mice were orally challenged with the adherent-invasive *E. coli* (AIEC) strain LF82 (once) and euthanized at day 7 after the challenge. Pair-feeding was performed during the first 12 weeks.

**Figure 7 ijms-23-05891-f007:**
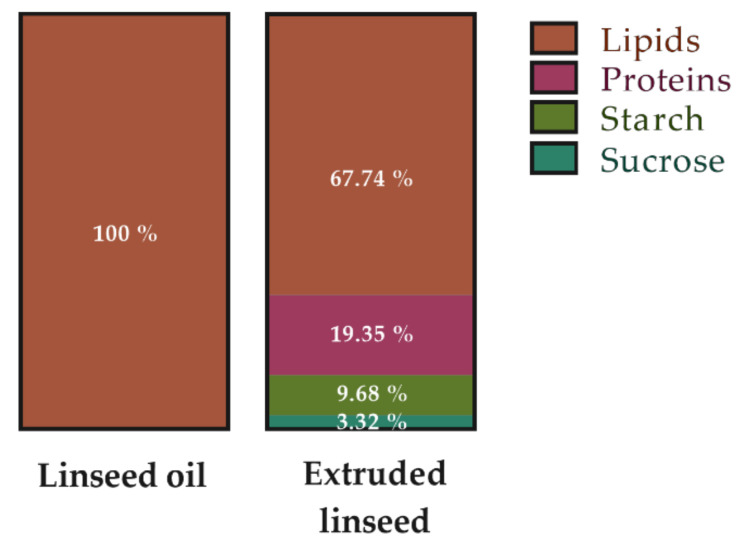
Macronutrient composition (%energy/100 g) of linseed oil (Vigean^®^, Clion, France) and extruded linseed (Tradi-Mega; Tradilin, Patent n°FR1760984) used as supplements.

**Table 1 ijms-23-05891-t001:** Fecal acetate, butyrate, and propionate (µmol·g^−1^ feces).

	HFD (*n* = 16)	HFD-LS-O (*n* = 16)	HFD-LS-E (*n* = 16)	*p*
Acetate	10.6 ± 4.7	11.2 ± 5.9	13.5 ± 4.4	*0.11*
Butyrate	0.18 ± 0.3	0.16 ± 0.2	0.35 ± 0.3	*0.13*
Propionate	2.22 ± 1.2	2.1 ± 0.7	2.7 ± 0.7	*0.07*
Total SCFAs	12.8 ± 5.8	13.3 ± 6.5	16.2 ±4.8	*0.09*

HFD (high-fat diet, n = 16), HFD-LS-O (linseed oil-supplemented HFD, n = 16), HFD-LS-E (extruded linseed-supplemented HFD, n = 16). SCFAs: short chain fatty acids; Total SCFAs: acetate + butyrate + propionate.

**Table 2 ijms-23-05891-t002:** Macronutrients (% kcal/100 g) in the three high-fat diets supplemented with linseed (HFD-LS-O and HFD-LS-E) or not (HFD) and their n-6/n-3 PUFAs ratio.

	HFD (*n* = 16)	HFD-LS-O (*n* = 16)	HFD-LS-E (*n* = 16)
Proteins (%kcal)	19.0	19.0	19.5
Carbohydrates (%kcal)	40.6	40.6	40.1
Lipids (%kcal)	40.4	40.4	40.4
SFAs (% of lipid fraction)	40.0	40.0	36.7
MUFAs (%)	44.1	44.3	41.0
PUFAs (%)	15.8	15.8	20.6
LA (C18:2 n-6) (%)	1.0	3.6	5.2
ALA (C18:3 n-3) (%)	14.8	12.1	15.4
kcal/100 g	468.4	468.4	460.4
n-6/n-3 PUFA ratio	14.8	3.3	3.0

HFD (high-fat diet, n = 16), HFD-LS-O (linseed oil-supplemented HFD, n = 16), and HFD-LS-E (extruded linseed-supplemented HFD, n = 16). SFAs: saturated fatty acids; MUFAs: monounsaturated fatty acids; PUFAs: polyunsaturated fatty acids; LA: linoleic acid; ALA: α-linolenic acid.

**Table 3 ijms-23-05891-t003:** Diet composition.

	HFD (*n* = 16)		HFD-LS-O (*n* = 16)		HFD-LS-E (*n* = 16)	
	*w*/*w* (%)	kcal/100 g	*w*/*w* (%)	kcal/100 g	*w*/*w* (%)	kcal/100 g
Casein	22	88	22	88	21	84
L-Cystine	0.3	1.2	0.3	1.2	0.3	1.2
Sunflower oil	2	18	0.99	8.91	2	18
Lard	19	171	19	171	17	153
Linseed oil			1.01	9.09		
Extruded linseed					6	15.12
Starch	17	68	17	68	15	60
Maltodextrin	7	28	7	28	7	28
Sucrose	22.55	90.2	22.55	90.2	22	88
Cellulose	5		5		4.55	
AIN93M	4		4		4	
AIN93Vx	1	4	1	4	1	4
Choline	0.15		0.15		0.15	
Total	100	468.4	100	468.4	100	460.44

HFD (high-fat diet, n = 16), HFD-LS-O (linseed oil-supplemented HFD, n = 16), and HFD-LS-E (extruded linseed-supplemented HFD, n = 16).

## Data Availability

Unprocessed 16S sequencing data are deposited in the European Nucleotide Archive under accession numbers PRJEB48648.

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
