# Peer review of "Beneficial Effects of Linseed Supplementation on Gut Mucosa-Associated Microbiota in a Physically Active Mouse Model of Crohn’s Disease"

_ijms, 2022, doi:10.3390/ijms23115891_

Round 1
Reviewer 1 Report
Plissonneau C et al. reported beneficial effects of linseed supplementation on gut inflammation and mucosa-associated microbiota in an active mouse model of Crohn´s Disease. In spite of the fact that the study is very relevant, the conclusions raised by authors are not well-supported with the results that they present in this manuscript.
For this reviewer, the protocol they have performed is not correct, since they state that supplementation of linseed is beneficial but they have not seen any induction of intestinal inflammation. In Figure 3 where they show the weight change, the three groups of mice exhibit the same change and the same intestinal barrier function. In addition, authors have not analysed any intestinal parameter such as MPO or gene expression of proinflammatory cytokines. Why suthors sacrificed the mice 7 days after the bacterial exposure? They should have waited more days in order to induce a higher intestinal inflammation and only in those conditions, authors could analyse the relevance of the lindseed supplementation in intestinal inflammation. Therefore, authors cannot conclude that this compound has a beneficial effect in intestinal inflammation.
On the other hand, authors have analysed the mucosa-associated microbiota in the colon but they have quantified the SCFAs levels in fecal samples. What about the levels of these metabolites specifically in the colon? Those levels could better correlate with the mucosa-associated microbiota.
Taking together, authors must change the protocol of colitis induction whether they want to demonstrate the beneficial effect of linseed in intestinal inflammation.
Reviewer 2 Report
This manuscript by Plissonneau et al. used linseed oil or extruded linseed supplementation to rebalance the dietary n-6/n-3 PUFA ratio of western diet and evaluated how these changes in diet affect colon inflammation. The authors found that adding extruded linseed to western diet alters mucosal microbiota composition. Specifically, the abundance of butyrate producing Firmicutes was increased, explaining the beneficial effect of extruded linseed in gut inflammation. Overall, the manuscript was very well organized, and the data provided support the main message. However, addressing the following points could further improve the prudence of the paper.
- In Figure 3A, the mice between three different group showed similar weight loss at day 1, 3, 6 after challenge. Did the author also monitor the weight loss during the recovery phase? Was any difference observed during this phase?
- Between line 155-158, the authors described the negative correlation between fecal lipocalin-2 concentration and the level of physical activity and hypothesized that the physical activity protects the mice from intestinal inflammation. However, is it possible that the mice without intestinal inflammation are more physically active simply because they are not as sick?
- The author challenged the mice with AIEC to model CD and sequenced the mucosal microbiota of mice fed with different diet. Did the author check how these diets affect the colonization of AIEC in the mucosal layer of the gut?
- The author should add more background information of the CEABAC10 transgenic mice in the introduction as this is the model they use throughout the paper. The will help the read to understand the rationale of their experimental design.
Reviewer 3 Report
In introduction section, please add several sentences to explain why select linseed.
All microbial names (e.g. Clostridiales spp in line 205) should be italic.
In M&M, please give a more clear description of experimental design.
